# Exome Sequencing Identifies a Biallelic GALNS Variant (p.Asp233Asn) Causing Mucopolysaccharidosis Type IVA in a Pakistani Consanguineous Family

**DOI:** 10.3390/genes13101743

**Published:** 2022-09-27

**Authors:** Saima Ghafoor, Karina da Costa Silveira, Raheel Qamar, Maleeha Azam, Peter Kannu

**Affiliations:** 1Translational Genomics Laboratory, COMSATS University Islamabad, Islamabad 45550, Pakistan; 2Department of Medical Genetics, University of Alberta, Edmonton, AB T6G 2H7, Canada; 3Pakistan Academy of Sciences, Islamabad 44000, Pakistan; 4Science and Technology Sector, ICESCO, Rabat 10104, Morocco

**Keywords:** mucopolysaccharidosis type IVA, GALNS, Pakistani consanguineous family, homozygous mutation

## Abstract

Mucopolysaccharidoses (MPS) type IVA is a lysosomal storage disease that mainly affects the skeletal system and is caused by a deficiency of the enzyme N-acetylgalactosamine-6-sulfatase (GALNS). The condition can mistakenly be diagnosed as a primary skeletal dysplasia such as spondylo-epiphyseal dysplasia, which shares many similar phenotypic features. Here, we utilised whole exome sequencing to make the diagnosis of MPS IVA in a resource poor country. We report for the first time the identification of a biallelic GALNS missense variant (c.697G>A, p.Asp233Asn) in the Pakistani population and highlight the potential contribution that academic institutions can make in rare disease diagnosis in the absence of a developed clinical genetic service.

## 1. Introduction

Mucopolysaccharidoses (MPS) are inherited metabolic conditions caused by enzyme deficiency or malfunction leading to an accumulation of lysosomal glycosaminoglycans (GAGs) [1]. GAGs are long chain carbohydrates which play an important role in the formation of bone, cartilage, tendons, skin, and connective tissues. The accumulation of lysosomal GAGs leads to cellular damage and abnormalities of the skeletal system [2].

The prevalence of MPS is 1/20,000 live births worldwide [1]. Of the seven different identified sub-types (MPS I, II, III, IV, VI, VII, and IX), autosomal recessive MPS IV (also called Morquio syndrome) primarily affects the skeleton with cognitive functioning largely spared. The incidence of MSP IV ranges from 1/40,000 to 1/200,000 births worldwide, but the prevalence in Pakistan is not known. MSP IV is subdivided into type IVA (95% of affected individuals) and IVB (5% of affected individuals) [3]. MPS IVA results from an alteration of N-acetylgalactosamine-6-sulfate sulfatase (GALNS) (EC 3.1.6.4) enzyme activity secondary to biallelic pathogenic variants in *GALNS* leading to the accumulation of keratan sulfate (KS) and chondroitin-6-sulfate (C6S) in various tissues, resulting in characteristic skeletal and connective tissues abnormalities [4]. The diagnostic radiographic changes of MPS IV includes dysostosis multiplex specifically odontoid hypoplasia which is a poorly developed connections in the first and second vertebrae of the neck (cervical spine), and platyspondyly, that lead to clinical complaints such as headaches, limited neck movement and pain in neck muscles [5,6,7]. Here, we describe the identification of a *GALNS* pathogenic variant in a Pakistani consanguineous family affected by short stature and a skeletal dysplasia. Biochemical diagnosis facilities were not accessible to the family. This case demonstrates the utility of whole exome sequencing in achieving the diagnosis of an orphan disease in a resource-poor country, thus facilitating reproductive risk counselling.

## 2. Methods

### 2.1. Ethical Approval and Patient Consent

This study was approved by the Ethics Review Board (ERB) of the Department of Biosciences, COMSATS University Islamabad, Pakistan, under notification no. CUI-Reg/Notif-658/19/685 dated 1st March 2019. The family was verbally informed about the aim of the study and written consent was obtained from the family prior to sample collection.

### 2.2. Sample Collection and DNA Extraction

Blood samples from the proband (V:3), an affected member (V:4) and unaffected available family members (IV:1, IV:2, V:1, V:3, V:4, V:6) (Figure 1) were collected in vacutainers containing ethylenediaminetetraacetic acid (EDTA). From the collected whole blood, DNA was extracted by the phenol/chloroform method [8].

### 2.3. Whole Exome Sequencing

Whole Exome Sequencing (WES) of the proband (V:3) was performed by LC Sciences (Houston, TX, USA), on the Illumina NovaSeq 6000 Sequencing System (PE150, 10 GB data per sample) using the Library Truseq DNA library and TruSeq DNA Exome, Platform Novaseq. The SureSelect Human All Exon V6 kit was used to capture the whole exome.

The data were processed at the data generation site. Briefly, as follows, reads were mapped to the hg19 human reference genome assembly and cleaned using a burrow wheeler aligner (BWA). Duplicate reads were removed using Picard tools and variant calling was generated with GATK HaplotypeCaller or UnifiedGenotyper. Processed data were returned to the research team for further informatics and biological impact analysis. Functional annotation of the detected variants was performed using myPhenoDB and Franklin. The myPhenoDB is a freely available database and repository used for interpretation of whole exome/genome sequencing data. The researchers can store the phenotype information, diagnosis and pedigree structure using myPhenoDB and then analyse exome VCF files from a single individual/families/cohort suspected of having a Mendelian disorder. The output spreadsheet provided the information related to variants in genes, which are then filtered according to the phenotype of the patient [9] (https://phenodb.org/ (accessed on 10 December 2021)). The Franklin software was used to detect variants related to rare diseases and their carrier screening. The data were uploaded as VCF/FASTQ files. Franklin also provides information regarding reported literature about known variants, as well as single nucleotide polymorphisms (SNPs) and copy number variations (CNVs) [10] (https://franklin.genoox.com/clinical-db/home (accessed on 10 December 2021)).

### 2.4. Data Analysis and Filtration

The data were next analyzed using an in-house pipeline, where the variants which had less than 1% frequency were identified using Genome Aggregation Database (GnomAD). GnomAD contains the information of 15,708 whole genomes and 125,748 whole exome data of different worldwide populations. The GnomAD mutations’ catalog is used for pathogenicity prediction of the identified variants. The input required in gnomAD is the gene name, mutation type and chromosomal region of the mutation. The output file provides information related to the mutation, including its pathogenicity and frequency [11] (https://gnomad.broadinstitute.org/ (accessed on 15 April 2022)). Pathogenicity of the identified mutations was also checked using MutationTaster with the option to prioritize trans homozygous inheritance pattern of the disorder. MutationTaster is a tool that detects the effect of a mutation using variable prediction models and provides high accuracy as it is interlinked with GnomAD. Besides MutationTaster, VCF files were also uploaded to MutationDistiller, a search engine which is integrated with MutationTaster, generating the predicted effect of the mutation [12] (https://www.mutationtaster.org/ (accessed on 15 April 2022). For the effect of the mutation on the protein structure, HOPE was used. HOPE analysis requires the input sequence, or Uniprot accession code, which is P34059 for GALNS, which selects the wild type amino acid. The change was then indicated to generate the predicted effect on the protein [13] (https://www3.cmbi.umcn.nl/hope/ (accessed on 20 April 2022)). HOPE was helpful in characterizing the biochemical effect of amino acid change from wild type to mutant as well as the effect of the amino acid change on the three-dimensional (3D) structure of the protein along with its interactions with other proteins.

### 2.5. Sanger Sequencing

Sanger sequencing for the confirmation of the candidate gene variant and for determination of segregation in the family was performed using PCR primers for *GALNS* (c.697G>A, p.Asp233Asn) 5′-GGAAGCCAACCTCACCCAGATC-3′ (forward) and 5′-GCTCTGTCCTTCATAAGCCACA-3′ (reverse). Touchdown PCR was performed from 50–60 °C and the amplification of PCR products were confirmed by gel electrophoresis. Sequencing was done with an automated DNA sequencing platform (ABI 3730 DNA Analyzer-96-well block). Results were analyzed using CodonCode Aligner (https://www.codoncode.com/aligner/ (accessed on10 December 2021)), which traced the variants in the sequencing reaction generated from genomic PCR. ABI files were uploaded that were aligned with the reference genomic sequence in text format to analyze the mutation in the family (https://www.codoncode.com/aligner/ (accessed on 10 December 2021)).

## 3. Results

### 3.1. Clinical Findings

A consanguineous family with three affected (V:3, V:4, V:5) and three unaffected (V:1, V:2, V:6) siblings was identified from Baluchistan, Pakistan with a provisional diagnosis of spondyloepiphyseal dysplasia. The physical characteristics of all the affected members were similar, with short stature, pectus carinatum, short neck (Figure 2a), valgus deformity of the knees (Figure 2b) and scoliosis (Figure 2c). Distinctive morphological features included frontal bossing, a flat nasal bridge, eversion deformity and curvature of the spine (Figure 2a–c). A waddling gait was present in all affected individuals. The radiographs of proband (V:3) showed short stubby phalanges, proximal pointing of the metacarpals, carpal bone delay (Figure 3a), bowing of the radius associated with mesomelic shortening (Figure 3b), hypoplastic femoral epiphyses with metaphyseal abnormalities of the knees (Figure 3c), sloping of tibia (Figure 3d), curvature of the spine, and platyspondyly with beaking of the vertebral bodies (Figure 3e).

### 3.2. Genetic Findings

WES analysis of the proband (V:3, Figure 1) resulted in the identification of a biallelic variant (c.697G>A, p.Asp233Asn) in *GALNS*, which has been previously shown to cause MPS type IVA. Sanger sequencing was performed to confirm the segregation in the family, revealing another affected sibling (V:4) to be also homozygous for this variant and the parents (IV:1 and IV:2) as carriers. Among the two unaffected siblings (V:1 and V:6) who were analysed, individual (V:1) did not carry the variant, while individual (V:6) was heterozygous. The affected sibling (V:5) was not tested due to non-availability of a DNA sample (Figure 4).

### 3.3. In Silico Protein Analysis

HOPE analysis showed that this variant was localized to the catalytic domain of the protein that is vital for enzyme activity. The catalytic domain is further connected to the N-terminal domain, which is essential for enzyme activity as it contains the active site at the centre of α/β domain. HOPE analysis further predicted that the variant may lead to a disruption in interaction between the two domains thus affecting enzyme function by affecting its active site that is suitable for binding polyanionic substrates such as keratan KS and C6S. The change from aspartic acid in the ancestral protein to asparagine in the variant protein causes a change in protein charge from negative to neutral, which potentially disturbs the ionic interaction of the enzyme with its ligands (Figure 5).

## 4. Discussion

Genomics is paving the way for an improvement in public health outcomes, including the early diagnosis and treatment of orphan diseases. MPS type IVA, also known as Morquio (Morquio-Brailsford) syndrome results from the accumulation of KS and C6S throughout the body affecting multiple organs and tissues including bones, cartilage, tendon, teeth, lungs, heart, cornea, skin and connective tissues [14]. Here, we report for the first time a Pakistani sibship affected by MPS IVA due to a biallelic variant (p.Asp233Asn) in *GALNS*. Since KS and C6S are stored primarily in the cartilage extracellular matrix, skeletal development is significantly affected in MPS IVA. Radiographs of the affected siblings suggested a provisional diagnosis of a spondyloepiphyseal dysplasia. Although certain characteristics such as proximal pointing of the metacarpals and vertebral beaking were suspicious of a storage disorder, biochemical testing for this group of disorders was not accessible or available to the family. In such a scenario where the differential diagnosis was wide open and there is no other method available to facilitate a precise diagnosis, WES provides a cost-effective strategy to achieve diagnosis. After the identification of the variant, its segregation with an abnormal phenotype was confirmed through Sanger sequencing. The family has been provided genetic counselling regarding the autosomal recessive nature of the condition and carrier risks. Carrier testing is now available to all family members via Sanger sequencing, which will be provided to them free of cost by us.

A report on 90 patients with MPS in Pakistan indicates the most frequent form of MPS is MPSI (83.33%) followed by MPS IVA (6.67%), MPS III (5.56%), MPS VI (3.33%), and MPS II (1.11%). Features commonly seen in this group of disorders includes a coarse facies, splenomegaly and characteristic skeletal manifestations [15]. Ullah (2017) performed the first molecular diagnosis of 8 Pakistani families affected by MPS and identified 4 novel variants p.Phe216Ser, p.Met38Arg, p.Ala291Ser, p.Glu121Argfs*37 and 2 recurrent variants p.Pro420Arg, p.Arg386Cys in *GALNS* [16]. Variant analysis in a different South Asian population from India has identified 40 different variants in *GALNS* (22 novel, 17 missense, 3 splice site, 1 nonsense and 1 frame shift mutation) [17]. Worldwide, more than 334 variants have been identified in GALNS including 203 missense and nonsense, 42 indels, 22 slice site alterations and 3 rearrangements [18,19,20]. The details of reported patients with *GALNS* mutation (p.Asp233Asn) are listed in Table 1 [18,19,20,21]. The full variant spectrum in South Asian countries including Pakistan, India and Bangladesh is yet to be determined since the cost of genetic testing remains prohibitive to most families [17]. 

Lysosomal storage diseases such as MPS IV can be treated with either enzyme replacement therapy (ERT), pharmaceutical chaperones, substrate reduction, or hematopoietic stem cell transplantation (HSCT) [25,26,27]. In ERT the patients are treated with recombinant enzymes approved by US Food and Drug Administration (FDA). The FDA has approved recombinant GALNS enzymes include elosulfase alfa and recombinant GALNS enzyme (rhGALNS) for the treatment of MPSIV. ERT is generally considered safe to use [28], however there are studies, which show that the ERT may not be able to completely alter the skeletal phenotype seen in this condition [25]. As metabolic disorders are congenital and associated with irreversible damage to the cells and tissues such as the bones, such as the bones, the physical manifestations may be too advanced at the time of clinical diagnosis to be reversed by the treatment. A timely and an accurate diagnosis of MPS IVA followed by ERT has been reported to show improvement in mobility as measured by the 6-min walking test [26]. HSCT on the other hand is a comparatively new treatment showing durable and normal activity of the enzyme, which has been reported to increase the density of lumber bone mineral, improved movement and reduction of narrow airways. However, this procedure is highly risky having multiple complications and an increased mortality rate [27,29,30]. Pharmacological chaperones are also being used for MPSIVA treatment [25]. These small molecules bind to the protein target and stabilize the protein conformation or promote the correct folding and trafficking of the altered protein. These molecules basically bind to the proteins active site increasing their thermal stability. The pharmacological chaperones can be administered orally and can be used as a monotherapy. Molecular chaperons have been reported to enhance the enzyme activity of the mutated protein and when coupled with ERT can increase the recombinant lysosomal enzymes activity, which suggests that these chaperon along with recombinant enzyme when co-administered can increase ERT efficacy [25]. The combination of both treatments is reported to reduce the lysosomal mass in the patient cells by the administration of hrGALNS with pharmaceutical chaperone (pranlukask) in MPSIVA fibroblast has been reported to reduce patient cell lysosomal mass and normalised cell function [29].

The disease burden and complications of MPSIVA include reduced mobility, the severity of which can lead to the use of wheelchair and pain affecting the spine and joints. Several psychological complications have also been noticed in patients including anxiety. Therefore, early diagnosis can be helpful in disease management. Spinal surgeries are often required to correct deformity, improve physical function and reduce pain [31]. An early diagnosis program has been proposed in Spain to detect the disease before the onset of irreversible and severe manifestations. The program also includes creating public awareness about MPS diseases throughout the country [32].

The necessity of instituting early treatment for lysosomal storage diseases such as MPS IV has led to the development of newborn screening methods to achieve rapid diagnosis after birth. However, the impact of early treatment following newborn screening remains to be determined and whether treatment will normalize skeletal development remains unknown. Presently, newborn screening for MPS in not available in Pakistan, in fact, Pakistan lacks a national program for newborn screening due to the lack of a robust infrastructure for health care delivery. Access to screening is primarily available on a user pays basis at a local level from private laboratories within Pakistan and abroad for specific conditions [33].

About 20 years ago, the World Health Organization (WHO) published its position on genomics and world health, including issues pertaining to the integration of genomics into healthcare, ethical issues, and the equitable sharing of benefits between developed and developing countries. Public health genomic policies are now being actively expanded in several resource rich countries worldwide. In addition, the number of orphan diseases screened is expected to increase as new treatments become available, shifting the imperative to early diagnosis prior to the development of symptom. Nevertheless, genetic services remain limited or unavailable to most, particularly in resource poor countries such as South Asia. Pakistan continues to be challenged in its ability to deal with public health issues stemming from infectious diseases and malnutrition, leaving limited resources available for medical genetic services and the development of clinical genomics expertise. Furthermore, the majority of births in Pakistan occur at home and there is limited genomic literacy in both the general population and health care professionals to handle postnatal screening [34]. According to the WHO, 70% of birth defects globally could potentially be prevented or treated if clinical genetic services in developing countries are adequately strengthened [35]. To further exacerbate the issue, the College of Physicians and Surgeons of Pakistan does not offer training in clinical genetics and graduate level genetic counsellor training is also unavailable in Pakistan.

The tertiary education system in Pakistan is mature and has for many years graduated students in biotechnology and molecular genetics. Furthermore, researchers in Pakistan generate high-quality genetic research, despite limited funding and medical infrastructure. We conclude that the current case highlights how the research community can assist in facilitating a medical diagnosis in the absence of a clinical genetics’ workforce. Unbiased sequencing technique such as whole exome sequencing has the ability to facilitate the rapid diagnosis of orphan diseases in a resource poor country, enabling the provision of accurate reproductive risk counselling.

## 5. Conclusions

Although MPS is a multisystem condition, MPS IVA can have predominant or isolated presentation of musculoskeletal involvement. Spondyloepiphyseal dysplasia or other basic musculoskeletal illnesses are frequently misdiagnosed as MPS IVA. Correct early diagnosis is essential to provide accurate management. In this study we performed whole exome sequencing for genetic-based accurate diagnosis of MPS in a Pakistani family. The genotyping resulted in the characterization of MPS IVA. Until now reported treatments for MPS IVA such as, ERT and HSCT are available in developing countries including Pakistan. Genetic testing and counseling coupled with national awareness campaigns can be helpful in reducing the disease prevalence in the Pakistani populations.

## Figures and Tables

**Figure 1 genes-13-01743-f001:**
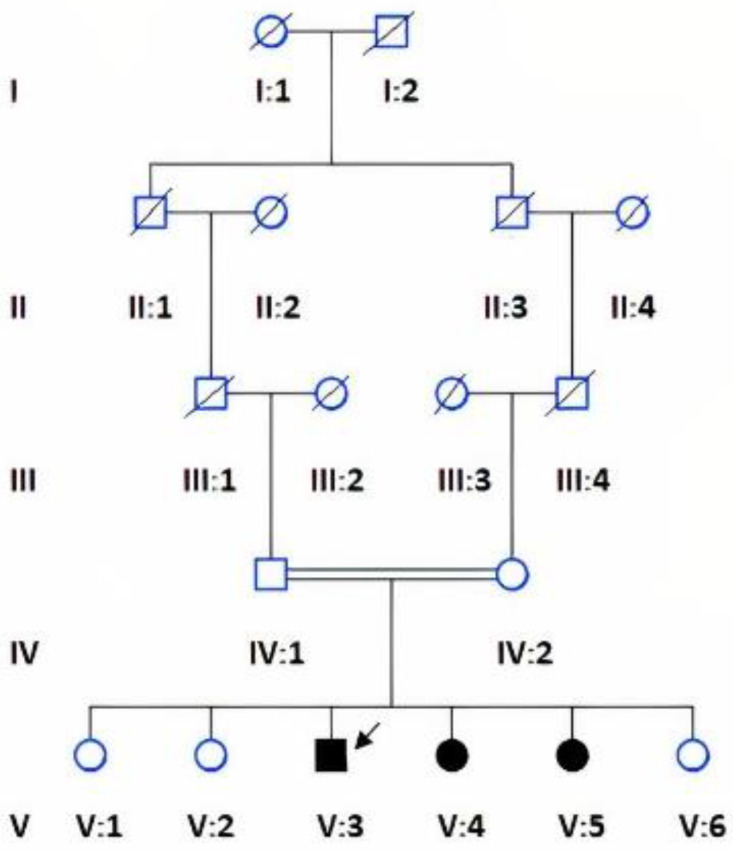
Pedigree of the family showing parental consanguinity, circles indicate female gender and squares males, filled square and circles indicate affected individuals, arrow points to the proband.

**Figure 2 genes-13-01743-f002:**
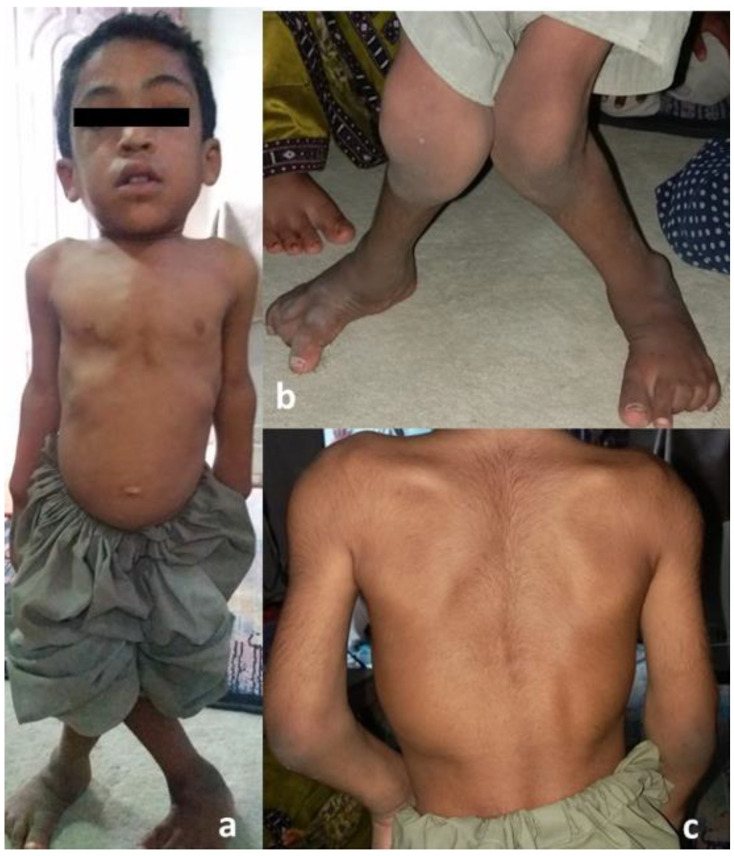
Morphological features of proband: (**a**) note the short stature, frontal bossing, flat nasal bridge, short neck, and protrusion of the chest.; (**b**) note the valgus knee deformity; and (**c**) curvature of spine/scoliosis.

**Figure 3 genes-13-01743-f003:**
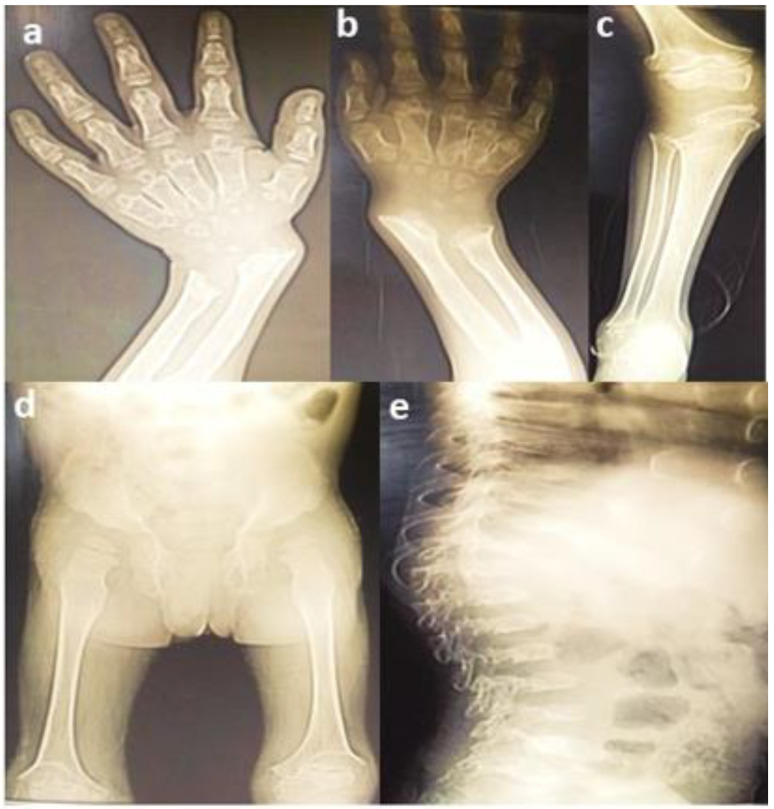
Radiographs of proband: (**a**) hand radiograph showing short phalanges, pointed metacarpals and underdeveloped carpal bones; (**b**) bowing of radius and mesomelia; (**c**) small femoral epiphyses, metaphyseal abnormality of the knees, flared distal ends of the femurs; (**d**) sloping of tibia; and (**e**) curvature of spine, short height vertebral bodies and platyspondyly with beaking of vertebral bodies.

**Figure 4 genes-13-01743-f004:**
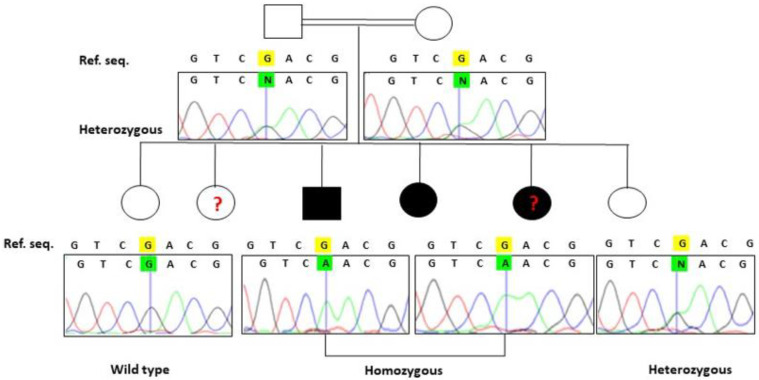
Sanger sequencing demonstrates variant segregation, “?” indicates that DNA were unavailable for analysis, highlighted in yellow ‘G’ is the wildtype nucleotide which is mutated to ‘A’ in patients and heterozygous members of the family.

**Figure 5 genes-13-01743-f005:**
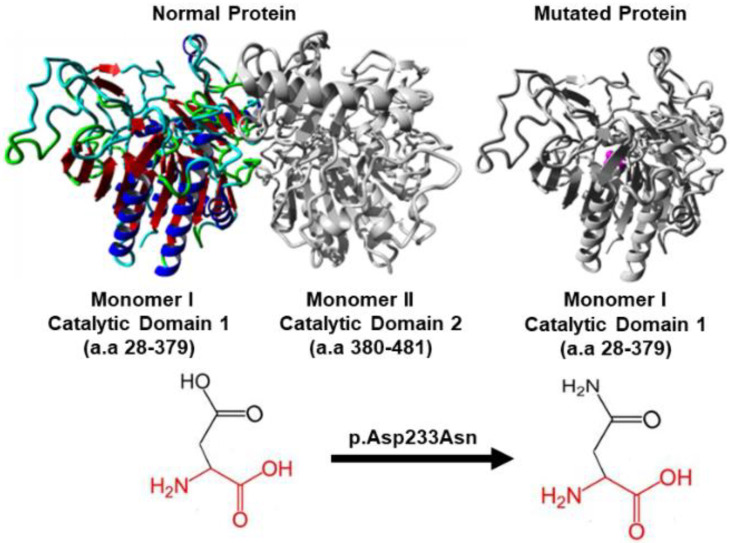
The p.Asp233Asn mutation analysis of GALNS using HOPE. The normal protein structure is represented as homodimer, with each monomers having catalytic domain. Domain 1 from amino acid (a.a) residues 28 to 379, and an antiparallel β-strands in domain 2 from a.a residues 380 to 481. The ancestral protein (**left** side) has aspartic acid (Asp) at position 233 and is negatively charged residing in domain 1, colored by element; α-helix = blue, β-strand = red, turn = green, 3/10 helix = yellow and random coil = cyan, while Domain 2 is colored grey. The monomer I with mutated residue (**right** side), shown in grey, has the side chain of asparagine (Asn) at position 233 (colored magenta) having neutral charge, is represented as small balls located at the catalytic domain 1 of the protein causing functional disturbance.

**Table 1 genes-13-01743-t001:** Current study and reported patients worldwide with *GALNS* mutation (p.Asp233Asn) where genotype GG is wild type, AA is homozygous mutant and GA is heterozygous.

Disease Severity	Mutation (p.Asp233Asn) Zygosity	Clinical Features of Patients	Mutated Allele (A) Count for c.697G>A (p.Asp233A)sn	Number of Individuals with Genotype	Allele Frequency Distribution	c.697G>A (p.Asp233Asn) Allele Frequency GnomAD	Study Type	Ethnicity	Reference
Severe	Homozygous (p.Asp233Asn)	Spondylo-epiphyseal dysplasia, short stature, chest protrusion, short neck, vulgus deformity of the knees, scoliosis, and, abnormal gait	7	GG = 2 AA = 4 GA = 3	16.6% 33.3% 50%	0.00001	Familial	Pakistani	Current study
Severe	Homozygous (p.Asp233Asn) Compound Heterozygous (p.Asp233Asn p.Ala296Val)	Bone dysplasia, short trunk, and corneal clouding	9	GG = 422 AA = 8 GA = 1	97.7% 1.9% 0.46%		Case study	Chinese	[22]
Severe	Compound heterozygous (p.Asp233Asn p.Lys153_Phe306del)	Spondyloepiphyseal dysplasia, chest deformity, coxa valga, hepatomegaly, hearing loss, and mental retardation	1	GG = 10 AA = 0 GA = 1	83.3% 0% 16.6%		Case study	Italian	[23]
Mild	Compound heterozygous (p.Gly168Arg) (Asp233Asn)	Chest deformity	1	GG = 114 AA = 0 GA = 1	98.2% 0% 1.7%		Familial	Polish	[24]
Severe	Homozygous (Asp233Asn)	Bone deformity, growth retardation, short stature, and cervical spine instability	8	GG = 488 AA = 8 GA = 0	98.3% 1.6% 0%		Familial	German, Polish, Indian	[17]

## Data Availability

Not applicable.

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
