# Peer review of "Exome Sequencing Identifies a Biallelic GALNS Variant (p.Asp233Asn) Causing Mucopolysaccharidosis Type IVA in a Pakistani Consanguineous Family"

_genes, 2022, doi:10.3390/genes13101743_

Round 1

Reviewer 1 Report

I received the article entitled “Exome Sequencing Identifies a Biallelic GALNS Variant 2
(p.Asp233Asn) Causing Mucopolysaccharidosis Type IVA 3 in a Pakistani Consanguineous Family IVA by Saima Ghafoor, Karina da Costa Silveira, Raheel Qamar, Maleeha Azam, and Peter Kannu

The authors describe the different types of mucopolysaccharidoses (MPS), focusing on type IVA; caused by a deficiency in the enzyme N-acetylgalactosamine-6-sulfatase. The authors utilize whole exome sequencing in order to make the diagnosis.  The authors report a biallelic GALNS missense variant in a Pakistani family.

              Overall the experimental observation that the authors have recognized is interesting and important.  I only have a few minor edit comments/changes:

Line 34               edit N-acetylgalactosamine-6-sulfatase (EC 3.1.6.4) – please add (EC 3.1.6.4)

Line 84               “60-50 oC” should read “50-60 oC” or “55 +/- 5 oC”

Line 124             Please define ‘HOPE’ and include a reference too

Line 175             Can the authors discuss more about treatment, it would add value to the manuscript if this section is expanded upon; Seems like a reference is needed here as well.

Line 233            References.  The authors do need to go through the references and make them consistent. For example, some of the references list the list author and then et al.  Others, all authors are listed.  The preferred way is to list all authors on each reference.  Please bold the year.

Author Response

MDPI Revision Reviewer 1

Comments and Suggestions for Authors

I received the article entitled “Exome Sequencing Identifies a Biallelic GALNS Variant 2
(p.Asp233Asn) Causing Mucopolysaccharidosis Type IVA 3 in a Pakistani Consanguineous Family IVA by Saima Ghafoor, Karina da Costa Silveira, Raheel Qamar, Maleeha Azam, and Peter Kannu

The authors describe the different types of mucopolysaccharidoses (MPS), focusing on type IVA; caused by a deficiency in the enzyme N-acetylgalactosamine-6-sulfatase. The authors utilize whole exome sequencing in order to make the diagnosis.  The authors report a biallelic GALNS missense variant in a Pakistani family.

Overall, the experimental observation that the authors have recognized is interesting and important.  I only have a few minor edit comments/changes:

Comment 1: Line 34: Edit N-acetylgalactosamine-6-sulfatase (EC 3.1.6.4) – please add (EC 3.1.6.4)

Answer: The required information has been added in “Introduction” paragraph 2 as “N-acetylgalactosamine-6-sulfatase (EC 3.1.6.4)”.

Comment 2: Line 84: “60-50 oC” should read “50-60 oC” or “55 +/- 5 oC”

Answer: Actually, we conducted a touchdown PCR in which we start at a higher temperature in this case 60 oC, decreasing the temperature in each cycle till we hit 50 oC that is why it has been written 60-50 oC. If there is any other conventionally used way of expressing this, it may be pointed out and we shall be happy to comply with it.

Comment 3: Line 124: Please define ‘HOPE’ and include a reference too.

Answer: The suggested information has been provided in “methods” sub-section “data analysis and filtration” in “methods” sub-section “Data Analysis and Filtration” as “For effect of mutation on the protein structure HOPE was used. HOPE analysis required the input sequence or Uniprot accession code which is P34059 for GALNS, which selects the wild type amino acid and the change was then indicated to generate the predicted effect on the protein (https://www3.cmbi.umcn.nl/hope/). HOPE was helpful in characterizing the biochemical effect of amino acid change from wild type to mutant as well as the effect of amino acid change on the three dimensional (3D) structure of the protein along with its interaction with other proteins”. The references are added as well.

Comment 4: Line 175: Can the authors discuss more about treatment, it would add value to the manuscript if this section were expanded upon; Seems like a reference is needed here as well.

Answer: As per suggestions of the reviewer 1, the details regarding the available treatments for MPS have been added  in section “Discussion” in paragraph 3 as“ In ERT the patients are treated with recombinant enzymes approved by US Food and Drug Administration (FDA) . The FDA approved recombinant GALNS enzymes include elosulfase alfa and recombinant GALNS enzyme (rhGALNS). ERT is generally considered safe to use as a treatment for MPSIV 28. However there are a few studies, which show that the ERT therapeutic effect in patients younger than 5 years of age didn’t exhibit considerable impact on the skeletal phenotype 25. As metabolic disorders are congenital and associated with irreversible damage to the cells and tissues such as the bones, thus the physical manifestations are devastating at the time of clinical diagnosis. A timely and an accurate diagnosis of MPS IVA followed by ERT has reported to show improvement in mobility as measured by the 6-minute walking test, however, treatment has not been shown to reverse the skeletal defects 26. HSCT on the other hand is a comparatively new treatment showing durable and normal activity of the enzyme, which has been reported to increase the density of lumber bone mineral, improved movement and reduction of narrow airways. However, this procedure is highly risky having multiple complications with increased mortality rate 27,29,30. Pharmacological chaperones are also being used for MPSIVA treatment 25. These small molecules bind to the protein target and stabilize the protein conformation or promote the correct folding and trafficking of the altered protein. These molecules basically bind to proteins active site increasing their thermal stability. The pharmacological chaperones can be administered orally and can be used as a monotherapy. Molecular chaperons have been reported to enhance the enzyme activity of the mutated protein and when coupled with ERT can increase the recombinant lysosomal enzymes activity, which suggests that these chaperon along with recombinant enzyme when administered can increase ERT efficacy25. The combination of both treatments is reported to reduce the lysosomal mass in the patient cells by administration of hrGALNS with pharmaceutical chaperone (pranlukask) in MPSIVA fibroblast. In addition, patients’ cells treated with the hrGALNS and pranlukast normalized the fibroblast up to the control cells level thus suggesting the use of both treatments together with improved efficacy of ERT”.

Additional citations are provided.

Comment 5: Line 233: References.  The authors do need to go through the references and make them consistent. For example, some of the references list the list author and then et al.  Others, all authors are listed.  The preferred way is to list all authors on each reference.  Please bold the year.

Answer: The references format has been corrected in the revised manuscript.

Thank you

Reviewer 2 Report

Dear Authors,

your paper points out many topic problems. The data are well detailed with appropriate references. Could you improve the description of the multiple dysostosis typical of MPSIV, especially the short neck with cervical spine deformity that lead to . Could you specify and explain better what is HOPE analysis? In the line 142 you cite KS but I never found any abbreviation before in the text. Could you detail better which alternative therapies (you cited chaperones and subtsrate reduction at line 174)for MPSIV?

I suggest to describe the compliocations of this disorder: neurosurgery, orthopedics, short stature and the therapeutical approches if an early diagnosis is made (as HSCT). 

I think this paper lays the foundation for the future with a polished analysis of cost/benefit and the health expenditure.

Author Response

MDPI Revision Reviewer 2

Comments and Suggestions for Authors

Dear Authors,

Your paper points out many topic problems. The data are well detailed with appropriate references. Could you improve the description of the multiple dysostosis typical of MPSIV, especially the short neck with cervical spine deformity that led to . Could you specify and explain better what is HOPE analysis? In the line 142 you cite KS but I never found any abbreviation before in the text. Could you detail better which alternative therapies (you cited chaperones and substrate reduction at line 174) for MPSIV?

I suggest to describe the complications of this disorder: neurosurgery, orthopedics, short stature and the therapeutical approaches if an early diagnosis is made (as HSCT). 

I think this paper lays the foundation for the future with a polished analysis of cost/benefit and the health expenditure.

Comment 1: Could you improve the description of the multiple dysostosis typical of MPSIV, especially the short neck with cervical spine deformity that led to.

Answer: As per suggestions by the reviewer 2, we have updated the description on page 1, paragraph 2 in "Introduction” section as “The radiographic changes characteristic of MPS IV are known as dysostosis multiplex, which include, platyspondyly, odontoid hypoplasia leading to clinical complaints such as headaches, limited cervical spine movement and neck muscle pain”.

Comment 2: Could you specify and explain better what is HOPE analysis?

Answer: As per suggestions of reviewer, the details of HOPE analysis are incorporated in the revised manuscript in “methods” sub-section “data analysis and filtration” as, “For effect of mutation on the protein structure HOPE was used. HOPE analysis required the input sequence or Uniprot accession code which is P34059 for GALNS, which selects the wild type amino acid and the change was then indicated to generate the predicted effect on the protein (https://www3.cmbi.umcn.nl/hope/). HOPE was helpful in characterizing the biochemical effect of amino acid change from wild type to mutant as well as the effect of amino acid change on the three dimensional (3D) structure of the protein along with its interaction with other proteins”

Comment 3: In the line 142 you cite KS but I never found any abbreviation before in the text.

Answer:  The mentioned abbreviation has been added in section “In Silico Protein Analysis” & lines 7 as “keratan sulfate (KS) and chondroitin-6-sulfate (C6S)”

Comment 4: Could you detail better which alternative therapies (you cited chaperones and substrate reduction at line 174) for MPSIV?

Answer: As per suggestions, the detail of available therapies have been provided in section “Discussion” in paragraph 3 as“ In ERT the patients are treated with recombinant enzymes approved by US Food and Drug Administration (FDA) . The FDA approved recombinant GALNS enzymes include elosulfase alfa and recombinant GALNS enzyme (rhGALNS). ERT is generally considered safe to use as a treatment for MPSIV 28. However there are a few studies, which show that the ERT therapeutic effect in patients younger than 5 years of age didn’t exhibit considerable impact on the skeletal phenotype 25. As metabolic disorders are congenital and associated with irreversible damage to the cells and tissues such as the bones, thus the physical manifestations are devastating at the time of clinical diagnosis. A timely and an accurate diagnosis of MPS IVA followed by ERT has reported to show improvement in mobility as measured by the 6-minute walking test, however, treatment has not been shown to reverse the skeletal defects 26. HSCT on the other hand is a comparatively new treatment showing durable and normal activity of the enzyme, which has been reported to increase the density of lumber bone mineral, improved movement and reduction of narrow airways. However, this procedure is highly risky having multiple complications with increased mortality rate 27,29,30. Pharmacological chaperones are also being used for MPSIVA treatment 25. These small molecules bind to the protein target and stabilize the protein conformation or promote the correct folding and trafficking of the altered protein. These molecules basically bind to proteins active site increasing their thermal stability. The pharmacological chaperones can be administered orally and can be used as a monotherapy. Molecular chaperons have been reported to enhance the enzyme activity of the mutated protein and when coupled with ERT can increase the recombinant lysosomal enzymes activity, which suggests that these chaperon along with recombinant enzyme when administered can increase ERT efficacy25. The combination of both treatments is reported to reduce the lysosomal mass in the patient cells by administration of hrGALNS with pharmaceutical chaperone (pranlukask) in MPSIVA fibroblast. In addition, patients’ cells treated with the hrGALNS and pranlukast normalized the fibroblast up to the control cells level thus suggesting the use of both treatments together with improved efficacy of ERT”.

Comment 5: I suggest to describe the complications of this disorder: neurosurgery, orthopedics, short stature and the therapeutical approaches if an early diagnosis is made (as HSCT). 

Answer: The suggested information has been now added in “Discussion” section paragraph 4 as “There is a variation in the spectrum of clinical manifestations experienced by individuals affected by a multisystem disease such as MPSIVA. Complications include a reduction in mobility leading to the use of a wheelchair. Joint pain is another significant problem. Cardiopulmonary disease secondary to valvular involvement and ventricular hypertrophy and spinal cord compression may further impact endurance and/or mobility. Quality of life is further compromised by vison and hearing impairments. Given the debilitation nature of MPSIVA, the cumulative disease burden affects the individuals psyche, leading to a loss of self esteem and depression. Orthopedic intervention is often required to prevent skeletal deformity, improve  physical function and reduce joint pain. Cervical spinal decompression and fusion surgery may be required to alleviate neural compression and stabilize the spine 31. In addition, programs have been proposed at the national level, with currently one successful program being that in Spain, which is focused on early diagnosis of MPS before the onset of irreversible and severe disease manifestations. The program also includes creating public awareness about MPS diseases throughout the country. Such model programs can be helpful if established worldwide”.

Thank you

Reviewer 3 Report

The paper by Ghafoor et al. describes the diagnosis of the pathogenic variant p.Asp233Asn in the gene encoding GALNS via exome sequencing in a consanguineous family from Pakistan. This genetic analysis confirmed the diagnosis of Mucopolysaccharidosis Type IV A (MPS IVA).

The authors claim that their methodological approach would be especially useful for diagnosing rare genetic diseases in resource poor countries with limited medical infrastructure.

Neither the method nor the identified p.Asp233Asn variant is new, but the paper shows how genetic diagnosis could still be performed in countries with limited medical infrastructure. In contrast to more economically developed countries, diagnosis of rare genetic diseases is very limited in countries such as Pakistan. Hence, many genetic diseases are under-diagnosed and founder-effects might not be identified. Therefore, a paper describing approaches to tackle this problem is worth publishing.

However, the manuscript should be improved and should better explain how to perform the data analysis with the internet sources that were used by the authors.

Specific comments:

Methods:

The authors made use of many publically available internet sources (e.g. Franklin, Gnomad, Mutation taster, HOPE), but don’t provide sufficient details how they used those sources. I recommend that the authors provide a more detailed description on how to use each of those sources and how to interpret the data. Are they all necessary? If not, please explain why.

As far as I know, ExAc is not available anymore?

Results:

Figure 3: there is no Figure 3E but a Figure 3F?

Figure 4: the letters are not readable, especially the ones highlighted in red.

Figure 5: I don’t understand this Figure. Figure 5A looks grey to me, but is described to be blue, red, green and yellow. Which structure refers to wildtype and which to the variant? What is shown in Figure 5B?

Discussion:

Line 168: Table 1 is mentioned to list worldwide GALNS mutations, but it only contains p.Asp233Asn variants.

Table1:

First row: I don’t understand the GG=7 from the current study. Did you sequence 7 individuals? As far as I understood from Figure 4, GALNS was sequenced in 6 individuals of whom one person is wildtype?

Allele Frequency: what is ExAC/TOPMED? What was calculated in this column?

Author Response

MDPI Revision Reviewer 3

Comments and Suggestions for Authors

The paper by Ghafoor et al. describes the diagnosis of the pathogenic variant p.Asp233Asn in the gene encoding GALNS via exome sequencing in a consanguineous family from Pakistan. This genetic analysis confirmed the diagnosis of Mucopolysaccharidosis Type IV A (MPS IVA).

The authors claim that their methodological approach would be especially useful for diagnosing rare genetic diseases in resource poor countries with limited medical infrastructure.

Neither the method nor the identified p.Asp233Asn variant is new, but the paper shows how genetic diagnosis could still be performed in countries with limited medical infrastructure. In contrast to more economically developed countries, diagnosis of rare genetic diseases is very limited in countries such as Pakistan. Hence, many genetic diseases are under-diagnosed and founder-effects might not be identified. Therefore, a paper describing approaches to tackle this problem is worth publishing.

However, the manuscript should be improved and should better explain how to perform the data analysis with the internet sources that were used by the authors.

Specific comments:

Comment 1:

Methods:

The authors made use of many publically available internet sources (e.g. Franklin, Gnomad, Mutation taster, HOPE), but don’t provide sufficient details how they used those sources. I recommend that the authors provide a more detailed description on how to use each of those sources and how to interpret the data. Are they all necessary? If not, please explain why.

As far as I know, ExAc is not available anymore?

Answer: As per suggestion of the reviewer, the details of Bioinformatic analysis are now included in the revised manuscript in “methods” sub-section “data analysis and filtration” as “Genome Aggregation Database (GnomAD). GnomAD contains information of 15,708 whole genomes and 125,748 whole exome data of different worldwide populations. The GnomAD mutations’ catalog is used for pathogenicity prediction of the identified variants. The input required in gnomAD is the gene name, mutation type and chromosomal region of the mutation. The output file  provides information related to the mutation including it’s pathogenicity and frequency11 (https://gnomad.broadinstitute.org/)”

As per suggestions, HOPE details are the details of HOPE analysis are incorporated in the revised manuscript in “methods” sub-section “data analysis and filtration” as, “For effect of mutation on the protein structure HOPE was used. HOPE analysis required the input sequence or Uniprot accession code which is P34059 for GALNS, which selects the wild type amino acid and the change was then indicated to generate the predicted effect on the protein (https://www3.cmbi.umcn.nl/hope/). HOPE was helpful in characterizing the biochemical effect of amino acid change from wild type to mutant as well as the effect of amino acid change on the three dimensional (3D) structure of the protein along with its interaction with other proteins”.

As per suggestions, MyphenoDB and Franklin description is also updated  in the “whole exome sequencing” sub-sections of the “methods”

The details of MutationTaster are provided in paragraph 2 in “data analysis and filtration” sub-section of “Methods” as “using myPhenoDB and Franklin. myPhenoDB is a freely available database and repository  used for interpretation of whole exome/genome sequencing data. The researchers can store the phenotype information, diagnosis and pedigree structure and analyse by using exome VCF files from a single individual/families/cohort suspected of having a Mendelian disorder. The output spreadsheet provided  the information related to variants in genes, which are then filtered according to the phenotype of the patient9 (https://phenodb.org/). The  Franklin software was used to detect variants related to rare diseases and their carrier screening. The  data were uploaded as VCF/FASTQ files. Franklin also provides information regarding reported literature about known variants, as well as single nucleotide polymorphisms (SNPs) and copy number variations (CNVs)”.

We do agree with the reviewer, as the ExAC is not available therefore ExAC as well as 1000Genome (an integrated database with ExAC) have been removed from the revised manuscript. An up-to-date database GnomAD (having the same utility as of ExAC and 1000Genome) is incorporated and the details of data analysis by GnomAD software have been incorporated in the revised manuscript. In addition, other online available databases for pathogenicity prediction of the mutations are also mentioned with details including, MutationTaster and MutationDistiller

Comment 2:

Results:

Figure 3: there is no Figure 3E but a Figure 3F?

Figure 4: the letters are not readable, especially the ones highlighted in red.

Figure 5: I don’t understand this Figure. Figure 5A looks grey to me, but is described to be blue, red, green and yellow. Which structure refers to wildtype and which to the variant? What is shown in Figure 5B?

Answer: The corrections have been incorporated in figure 3 with updated figure legends, as well as figure 4 with updated figure description and legends. In addition, Figure 5 is updated, and the figure legends are changed as mistakenly we had mentioned the wild type protein as mutant and mutant as wild type.

Comment 3:

Discussion:

Line 168: Table 1 is mentioned to list worldwide GALNS mutations, but it only contains p.Asp233Asn variants.

Table1:

First row: I don’t understand the GG=7 from the current study. Did you sequence 7 individuals? As far as I understood from Figure 4, GALNS was sequenced in 6 individuals of whom one person is wildtype?

Allele Frequency: what is ExAC/TOPMED? What was calculated in this column?

Answer: In the current study, in total 6 individuals were genotyped with 2 affected members with mutated allele (mutated allele “A” count=4), 3 heterozygous carriers (mutated allele “A” count=3) and one healthy member with homozygous wild type GG (normal allele “G” count=2). The following point have been clarified by correcting the allele count in column 3, and changing titles of the column 4-7 of  Table 1. In addition, the text in the discussion section has also been update in paragraph 2line 11 and 12 as “The details worldwide reported patients with GALNS mutation (p.Asp233Asn) are listed in Table 1”.

The correction of allele annotation has also been made G being wild type and A being mutated. The global minor allele frequency has been update as per GnomAD in Table 1.

Thank you

Round 2

Reviewer 3 Report

The manuscript has improved. My questions were answered.

However, Figure 5 is still not corrected. I think Figure 5 A is the mutant? Why does Figure 5 B contain two overlapping structures (one coloured one, one grey)? Is one from front and one from the back?

Author Response

Author Response Letter

The Reviewer

Subject: Submission of second round revisions of manuscript entitled “Exome Sequencing Identifies a Biallelic GALNS Variant (p.Asp233Asn) Causing Mucopolysaccharidosis Type IVA in a Pakistani Consanguineous Family” having manuscript ID : genes-1892203.

Thank you for your review of the above manuscript. Below we respond to the comment and indicate where changes have been made in the manuscript.

MDPI Revision Second Round Reviewer 3

Comments and Suggestions for Authors

The manuscript has improved. My questions were answered.

However, Figure 5 is still not corrected. I think Figure 5 A is the mutant? Why does Figure 5 B contain two overlapping structures (one coloured one, one grey)? Is one from front and one from the back?

Answer: The corrections in figure 5 are incorporated in the updated figure and figure legends on page no. 8 and in lines 188-201 as “The p.Asp233Asn mutation analysis of GALNS using HOPE. The normal protein structure is represented as homodimer, with each monomers having catalytic domain. Domain 1 (amino acid residues 28-379) and an antiparallel β-strands in domain 2 (amino acid residues 380-481). The ancestral protein (left side) has aspartic acid (Asp) at position 233 and is negatively charged residing in domain 1, colored by element; α-helix = blue, β-strand = red, turn = green, 3/10 helix = yellow and random coil = cyan, while Domain 2 is colored grey. The monomer I with mutated residue (right side), shown in grey, has the side chain of asparagine (Asn) at position 233 (colored magenta) having neutral charge, is represented as small balls located at the catalytic domain 1 of the protein causing functional disturbance”.

Maleeha Azam
